# Evaluation of the Anti-Cancer Potential of *Rosa damascena* Mill. Callus Extracts against the Human Colorectal Adenocarcinoma Cell Line

**DOI:** 10.3390/molecules27196241

**Published:** 2022-09-22

**Authors:** Hadeer Darwish, Sarah Alharthi, Radwa A. Mehanna, Samar S. Ibrahim, Mustafa A. Fawzy, Saqer S. Alotaibi, Sarah M. Albogami, Bander Albogami, Sedky H. A. Hassan, Ahmed Noureldeen

**Affiliations:** 1Department of Biotechnology, College of Science, Taif University, P.O. Box 11099, Taif 21944, Saudi Arabia; 2Department of Chemistry, College of Science, Taif University, P.O. Box 11099, Taif 21944, Saudi Arabia; 3Department of Physiology, Faculty of Medicine, Alexandria University, Alexandria P.O. 21544, Egypt; 4Center of Excellence for Research in Regenerative Medicine and Applications (CERRMA), Faculty of Medicine, Alexandria University, Alexandria P.O. 21544, Egypt; 5Department of Biology, College of Science, Taif University, P.O. Box 11099, Taif 21944, Saudi Arabia; 6Department of Biology, College of Science, Sultan Qaboos University, P.O. Box 36, Muscat 123, Oman

**Keywords:** *Rosa damascena*, callus induction, bio-elicitors, colorectal cancer cell line, anti-cancer activity

## Abstract

Chemotherapy is an aggressive form of chemical drug therapy aiming to destroy cancer cells. Adjuvant therapy may reduce hazards of chemotherapy and help in destroying these cells when obtained from natural products, such as medical plants. In this study, the potential therapeutic effect of *Rosa damascena* callus crude extract produced in vitamin-enhanced media is investigated on colorectal cancer cell line Caco-2. Two elicitors, i.e., L-ascorbic acid and citric acid at a concentration of 0.5 g/L were added to the callus induction medium. Callus extraction and the GC–MS analysis of methanolic crude extracts were also determined. Cytotoxicity, clonogenicity, proliferation and migration of Caco-2 colorectal cancer cells were investigated using MTT cytotoxicity, colony-forming, Ki-67 flow cytometry proliferation and Migration Scratch assays, respectively. Our results indicated that L-ascorbic acid treatment enhanced callus growth parameters and improved secondary metabolite contents. It showed the least IC50 value of 137 ug/mL compared to 237 ug/mL and 180 ug/mL in the citric acid-treated and control group. We can conclude that *R. damascena* callus elicited by L-ascorbic acid improved growth and secondary metabolite contents as well as having an efficient antiproliferative, anti-clonogenic and anti-migratory effect on Caco-2 cancer cells, thus, can be used as an adjuvant anti-cancer therapy.

## 1. Introduction

Colorectal cancer is the fourth frequently diagnosed cancer after lung, prostate and breast cancers [1]. In 2020, 104,610 new cases of colon cancer and 43,340 cases of rectal cancer were estimated to occur. During the same year, 53,200 people were estimated to die of colon and rectal cancer combined [2]. In the past 30 years, the mortality from colon cancer has reduced slightly due to early diagnosis [3]. However, the incidence of colon cancer has increased in people younger than 65 years old, with a 1% yearly rise in those aged between 50 to 64 years and 2% for those younger than 50 years old [1]. The risk of increasing colon cancer is influenced by genetic and acquired risk factors. Acquired risk factors associated with this disease include (i) dietary factors, for example low intake of fruit, fiber or vegetables, and high intake of red meat, caffeine, saturated fat and alcohol; (ii) lifestyle factors such as smoking and absence of exercise; (iii) side effects of some medical or surgical procedures, such as pelvic irradiation, cholecystectomy and ureterocolic anastomosis and (iv) co-morbid medical conditions such as human immunodeficiency virus infection, diabetes mellitus and inflammatory bowel disease [4]. Surgical resection is the best treatment for colon cancer but chemotherapy, radiation therapy and immunotherapy also play an important role in inhibiting recurrence and metastasis. Chemotherapy is the main therapeutic strategy in many incidences. It uses various drugs or combinations of drugs to diminish the cancer cell division. The classical route for delivering chemotherapy for colon cancer comprises delivering drugs to non-target positions; thus, patients suffer from side effects such as gastrointestinal toxicity, anemia, diarrhea, neutropenia, vomiting, mucositis, liver toxicity, hematologic disorders, damage to the nervous system and memory problems [5]. To date, the results of used treatments have not reached an acceptable outcome due to the risk of side effects and resistance to chemotherapy.

The use of medicinal plants as a natural source of pharmaceuticals has become increasingly popular in recent years. As a result of urbanization, overgrazing, pollution and the growth of agricultural regions, medicinal plants have been targeted for uncontrolled gathering and destruction. Secondary metabolism in plants produces a diverse range of chemically complicated chemicals, many of which are commercially valuable. Secondary metabolites are plant products of great pharmacological value, such as phenolics, terpenoids, glycosides, alkaloids and other compounds [6]. Secondary metabolites are often extracted from intact plants for commercial use. The three main chemical categories of secondary metabolites in plants are nitrogen compounds, terpenes and phenolic compounds [7]. Secondary metabolites that contain nitrogen have a basic nature and are identified by the presence of nitrogen in their fundamental structure. Alkaloids, glycosides and nonprotein amino acids are among plants’ most prevalent nitrogen-containing substances [8,9]. These metabolites are crucial for plants’ defense against insects and animals. The majority of alkaloids are extremely dangerous to humans, although small dosages of these substances may have medicinal benefits. Alkaloids or extracts containing alkaloids have been used as muscle relaxants, analgesics and tranquilizers from ancient times to the present [10]. The building block for terpenes, which are organic compounds, is an isoprene compound with five carbon atoms. Monoterpenes (C10), sesquiterpenes (C15), diterpenes (C20), triterpenes (C30), tetraterpenes (C40) and polyterpenes, with more than 40 carbons, are the different subgroups of terpenes [8,11]. Sesquiterpenes and volatile monoterpenes make up the majority of an essential oil’s chemical makeup [12,13]. Triterpenes, tetraterpenes and polyterpenes are the most prevalent terpenes [8]. Some terpenes, such as gibberellins, carotenoids and brassinosteroids, are also crucial for plant development in addition to their roles as anti-herbivore defensive chemicals in plants [9,14,15]. Phenolic chemicals are aromatic molecules made up of two groups: a hydroxyl (OH) group and a phenyl (C6) group. Simple molecules (such as phenolic acids) and highly polymerized molecules have different types of structures in these molecules (condensed tannins). Lignin, flavonoids, tannins, phenolic acids and coumarins are the five subgroups that make up phenolic chemicals [16]. The development of plant structure is affected by phenolic compounds, and they are also linked to a number of physiological functions, such as the defense against infections, insects and animals [17]. These substances are also recognized for their cardiovascular, gastrointestinal, antibacterial, antiviral, anti-cancer, anti-inflammatory, antiatherogenic, antithrombotic, analgesic and antithrombotic properties [18,19,20,21,22,23].

Plant cell culture is a renewable and environmentally acceptable source of secondary metabolites. Several findings show that plants can create a wide range of secondary metabolites, with some of them being commercially produced. Tissue culture methods have been created for a number of plants, but there are many others that are being over-exploited in the pharmaceutical industry and other disciplines, and need to be protected [24]. For generations, roses have been one of the most popular ornamental plants in the world. *Rosa damascena* Mill. (Damask rose) is one of the oldest and most valuable variants among the Rosaceae family. This rose is also used to make products with a variety of uses, including aromatherapy, antiseptic, antispasmodic, astringent agent, sedative, blood cholesterol altering, antibacterial, antimicrobial [25,26], anti-oxidizing [27] and anti-HIV effects [28]. *R. damascena* also has a variety of uses in the perfume, cosmetic and food sectors, such as the creation of rosewater, jam and dried flowers [29].

Vitamins may be conceived as bioregulator or hormone precursor chemicals that have a beneficial effect on plant growth and development when present in minute amounts. Therefore, these compounds might have an impact on the metabolic process for energy [30,31]. Ascorbic acid, a form of vitamin C, is more or less necessary for many vital physiological functions, including cell division, nutrition and water absorption, photosynthesis and the manufacture of enzymes and secondary metabolites. It serves as an enzyme cofactor, an antioxidant and a precursor for the production of oxalate and tartrate. Ascorbic acid is connected to chloroplasts that reduce the impact of oxidative stress during photosynthesis. Additionally, it prevents cell division from altering and functions as the main substrate in the cyclic pathway of hydrogen peroxide enzymatic detoxification [32].

The effect of these metabolites on cancer cell growth and behavior needs to be explored. Thus, the aim of this study is to investigate the potential effect of *R. damascena* callus elicited by two vitamins, ascorbic and citric acids, on cancer cells using colorectal cancer cell line Caco-2 as an in vitro model.

## 2. Results

### 2.1. R. damascena Callus Fresh, Dry and Crude Weight (g)

We looked for the suitable bio-elicitor to promote the productivity of phenolic content obtained from *R. damascena* callus’ fresh or dry weight in a series of early studies. This experiment’s outcomes in Table 1 and Figure 1A show that the ascorbic acid concentration of 0.5 g/L resulted in the highest fresh weight (2.878 g) and the maximum dry weight (0.306 g). The increase in fresh weight was not significant when compared to the control, but it was substantial in the callus’ dry weight when compared to the control and citric acid treatment. Table 1 also presents that the control treatment showed the biggest increase in crude weight (0.038 g). The presence of ascorbic acid or citric acid in the media resulted in a lower weight of crude output than the control treatment, with values of 0.032 g and 0.022 g, respectively, without significant differences between them.

### 2.2. GC–MS Analysis of R. damascena Callus

#### 2.2.1. Control Treatment

The crude yield of *R. damascena* callus in the control treatment was 0.038 g. Table 2 and Figure 1B represent the GC–MS analysis for *R. damascena* callus in the control treatment. The obtained result shows the presence of eight compounds in the chromatogram. The major compounds were 13.75% of Octadecanoic acid, methyl ester (CAS)—which was reported to be the highest composition of the compounds—followed by benzonitrile (CAS) with 9.15% and 1,2-Benzenedicarboxylic acid, bis(2-ethylhexyl) ester (CAS) with 3.05%. The 4-Octanol, propanoate, 1-Pentanol, 2,2-dimethyl-(CAS) and 2,6-Nonadien-1-ol had values of 1.55%, 1.60% and 1.74%, respectively, whereas (5,10,15,20-Tetraphenyl [2-(2)H1]prophyrinato)zinx (II) achieved the lowest amount (0.96%). The components of *R. damascena* crude differ according to the bio-elicitor callus they were exposed to.

#### 2.2.2. Citric Acid Treatment

Table 3 and Figure 1C represent the GC–MS analysis for *R. damascena* callus in the citric acid treatment. The obtained result shows the presence of eight compounds in the chromatogram. The major compounds included Octadecane, 2-methyl-—which is reported to be highest composition of the compounds (5.82%)—followed by Tetratetracontane (CAS) with 3.58% and 1,2-Benzenedicarboxylic acid, diisooctyl ester (CAS) with 2.69%. However, Tetradecane (CAS) and Dodecane 5,8-diethyl-(CAS) recorded 2.00% and 1.16%, respectively. Dichloroacetaldehyde valued 0.95%, whereas 3,3′,5,5′-Tetrabromo-2-nitro-2′-propylsulfonylbiphenyl and 5á-Pregnan-20-one, 3à,11á,17,21-tetrakis(trim ethylsiloxy)-, O-methyloxime revealed the lowest amounts—0.76% and 0.73%, respectively.

#### 2.2.3. L-Ascorbic Acid Treatment

Table 4 and Figure 1D represent the GC–MS analysis for *R. damascena* callus in the ascorbic acid treatment. The obtained result shows the presence of eight compounds in the chromatogram. The most prevalent compounds were 1,2-Benzenedicarboxylic acid, mono(2-ethylhexyl) ester (17.75%), 7-Octadecenoic acid, methyl ester (CAS) with 8.57% and Hexadecanoic acid, methyl ester (CAS) with 5.82%. Nonane, 1-chloro-(CAS) and11-[(t-Butyldimethylsilyl)oxy]-6,9a-dimethyl-6-(methoxycarbonyl)-(perhydro)napthaleno[a]benzofulvene valued 1.98% and 1.79%, respectively, whereas (2,3-Dihydro-5,10,15,20-tetraphenyl [2-(2)H1]prop hyrinato)copper(II) achieved the lowest amount (1.09%).

### 2.3. In Vitro Anti-Cancer Study (Cell Line Studies)

#### 2.3.1. Cellular Cytotoxicity Assay

Cytotoxicity of control and treated callus were investigated by an MTT assay. A concentration that kills 50% of the cells (IC50) was determined and used to compare the activity of different preparations. The IC50 values were 180.6 ug/mL for the control group, and 137.8 ug/mL and 237 ug/mL for the LAA- and CA-treated groups, respectively. The results prove that LAA treatment is the most potent cytotoxic one (Figure 2A–D).

#### 2.3.2. Clonogenic Assay

Digital images of the colonies were obtained using a camera, and colonies were counted for the calculation of plating efficiency for three culture dishes in each group in three separate experiments. The results revealed a significant decrease in the LAA-treated group compared to the untreated, control- and CA-treated groups. There was no significant difference between the control- and CA-treated groups, yet they were both significantly lower than the untreated one (Figure 2D–H).

#### 2.3.3. Ki-67 Flow Cytometry Proliferation Assay

A negative control of untreated Caco-2 cells was used to gate the Ki-67 negative population. The Ki-67 expression on untreated cells showed that 69.3% of cells were undergoing proliferation. Proliferation was significantly decreased in cells with the control-, LAA- and CA-treated callus, showing 38.4%, 31.7% and 40.4%, respectively. The control- and CA-treated cells showed no significant difference compared to each other. On the contrary, the LAA-treated cells group showed a significant reduction in cell proliferation in comparison to the CA and control groups (Figure 3).

#### 2.3.4. Migration Assay (Scratch Assay)

A comparison between untreated cells and the control-, LAA- and CA-treated groups was based on measuring the percentage of wound closure—a decrease in this percentage was taken as an indication of the decrease in cells’ migratory ability. Areas were quantified using Image J software (1.52p software 32, NIH, USA). The results showed that the least average wound closure percentage was for the LAA-treated group, with 17% after 24 h and 33% after 48 h. This was significantly less than that of the control- and CA-treated groups, showing 26% and 34% after 24 h, and 44% and 56% after 48 h, respectively (Figure 4).

## 3. Discussion

The components of *R. damascena* crude differ according to regions and the elicitors used in the experiments. Obviously, the results showed that ascorbic acid treatment recorded the highest values of fresh and dry weights of rose callus with a reliable value of crude weight when compared with the control treatment. Ascorbic acid is the major compound functioning in plant antioxidant systems [33]. It is implicated in cell division, cell elongation and synthesis of phytohormones [34]. Furthermore, it plays a key role in protecting plants from ROS through the ascorbate–glutathione cycle, as well as acting as a cofactor to violaxanthin de-epoxidase, which is considered to be an important enzyme in the photoprotective xanthophyll cycle during photosynthesis [35]. All of the previously mentioned positive effects of ascorbic acid can explain its significant improvement in the concentration of fresh, dry and crude weight under the circumstances of this study. This study evaluated the compounds obtained from the GC–MS analysis of *R. damascena* callus methanol extracts. Octadecanoic acid, methyl ester (CAS) is reported to be the major compound obtained from the GC–MS analysis of *R. damascena* callus in the control treatment, whereas (5,10,15,20-tetraphenyl [2-(2)H1]prophyrinato)zinx (II) recorded the lowest value. Our investigation was in line with those recorded by [36], who mentioned a significant presence of βcitronellol (30.24–31.15%); trans-geraniol (20.62–21.24%), n-heneicosane (8.79–9.05%), n-nonadecane (8.51–8.77%), nonadecene (4.42–4.55%) and phenylethyl alcohol (4.04–4.16%) was detected from the GC–MS chromatograph profile of *R. damascena* essential oil. GC–MS analysis for the methanol extract of *R. damascena* callus exposed to citric and ascorbic acids resulted in 16 compounds in the chromatogram, which confirmed that biotic elicitors increased the secondary products content. Obtained results of our study agreed with [37], who demonstrated that the exogenous application of ascorbic acid could enhance foliar growth, which may contribute to increased plant biomass and the yield of secondary products. Ascorbic acid was considered as an antioxidant involved in cell division and elongation [38]. Moreover, it also enhanced shoot formation in both young and old tobacco callus [39]. Furthermore, the application of ascorbic acid significantly improved photosynthetic pigments, gas exchange and nutrient content. This enhancement could be due to the functions of ascorbic acid as an essential cofactor of various enzymes or protein complexes [40,41]. This finding is in harmony with [42,43], who found that ascorbic acid foliar spray had a positive impact on total phenols, total chlorophyll, total carbs and percentages of N, P and K in lettuce. Ascorbic acid supplementation, in accordance with [44], enhanced N, P and K accumulation, and had a significant effect on nutrient uptake. Additionally, ascorbic acid enhanced the osmoprotectant proline and soluble carbohydrates, as well as the antioxidant enzymes CAT, POD, APX and SOD. This impact might be brought on by ascorbic acid, a water-soluble antioxidant that modulates a wide range of biological processes that influence plant development [45,46]. It is also involved in a variety of metabolic processes and has a dynamic connection with reactive oxygen species (ROS) [47,48,49]. In this regard, Sajid and Aftab [50] discovered that a foliar application of ascorbic acid significantly boosted the antioxidant enzyme activity of SOD, POD, CAT and APX in potatoes under salinity stress conditions. The variation in chemical composition between our research (greater 7-Octadecenoic acid, methyl ester (CAS) content and lower n-heptadecane content) and the presented data could be attributable to the bio-elicitors to which the plant was subjected.

Investigating the potential effect of the crude, ascorbic acid and citric acid-treated callus as an anti-cancerous adjuvant, human colorectal carcinoma cell line Caco-2 was employed in this study. The LAA-treated group was the most potent regarding cytotoxicity, where the IC50 was 137 ug/mL compared to 237 ug/mL and 180 ug/mL in the CA and control groups, respectively. This was further confirmed by the results of the clonogenic assay, where the colonies formed were the lowest in cells treated with LAA, denoting the lowest reproductive viability. The control- and CA-treated groups’ ability to produce progeny was also decreased in comparison to the untreated cells, but with no significance between them. Moreover, a negative control of untreated Caco-2 cells was used to gate the Ki-67 negative population. The Ki-67 status of Caco-2 cells treated with LAA showed the least proliferation capacity. Proliferation was also decreased in the CA-treated and control groups rather than the untreated group, suggesting that the anti-cancerous potential of *R. damascena* metabolites may be due to inhibiting the proliferation of cancer cells. In this study, a wound-healing scratch test was employed to reflect the migration potential of Caco-2 cells, and thus, their metastasizing ability. The results indicated that the ability of the LAA-treated group would significantly decrease the migration of Caco-2 cancer cells more than the CA-treated and control crude groups, which would result in better therapeutic outcomes.

## 4. Materials and Methods

This research was carried out in the Tissue Culture Laboratory of Taif University—where tissue culture techniques were used to increase the production of secondary metabolites in *R. damascena* (Family: Rosaceae)—and in the Center of Excellence for Research in Regenerative Medicine and its Applications (CERRMA) in Faculty of Medicine, Alexandria University—for the application of *R. damascena* callus crude extracts on cancer cell line Caco-2 to evaluate their potential anti-cancer effect. The current study was carried in three parts: the first was the induction of *R. damascena* callus; the second included an increment of secondary metabolites utilizing various elicitors, extraction, and GC–MS measurement; the third applied these metabolites on a cultured cancer colon cell line Caco-2 to evaluate their effect on cell viability, proliferation, clonogenicity and migration.

### 4.1. Callus Initiation of R. damascena

The goal of this section was to receive adequate quantities of callus. *R. damascena* mature closed flowering buds (1–1.5 cm long) were collected from the Al-hada district of Taif governorate for this investigation.

#### 4.1.1. Explant Sterilization

Closed mature flowering buds were washed in tap water with soap and a few drops of Tween-20 (a wetting agent that lowers the surface tension, enabling better surface contact), then submerged for 1 min in 70% ethanol, washed with sterile distilled water, and immersed for 5 min in a 5% marketing Clorox solution. To eradicate residuals, explants were disinfected three times with sterile distilled water in a laminar air flow hood.

#### 4.1.2. *R. damascena* Callus Generation in MS Medium Culture

For callus initiation, sterilized explants were grown in MS medium supplemented with 0.1 mg/L kinetin (Kin) and 1.0 mg/L naphthaleneacetic acid (NAA). Callus was obtained after 6 weeks of culture on medium.

### 4.2. R. damascena Elicitation Secondary Metabolites

Flowering callus buds were cultivated on Murashige and Skoog (MS) medium with citric acid at 0.5 g/L, L-ascorbic acid at 0.5 g/L, sucrose at 30 g/L, agar at 7 g/L, and produced from MS media supported with 0.1 mg/L kinetin (Kin) and 1.0 mg/L naphthaleneacetic acid (NAA).

#### 4.2.1. Experiment Design and Setting

The three following treatments were performed: 1-0.1 mg/L KIN and 1.0 mg/L NAA (control group), 2-L-ascorbic acid at 0.5 g/L (LAA group) and 3-citric acid at 0.5 g/L (CA group). Every three explants received ten jars in each treatment. Before autoclaving, the pH of the media was adjusted to 5.7–5.8 using a pH meter and an appropriate amount of 0.1 N HCl or 0.1 NaOH. Clean jars were used to deliver the material. Each one had 30 mL of nutritional media in it. After that, the jars were autoclaved for 15 min at 121 °C, 1.5 kg/cm^3^ at 1.5 kg/cm^3^. All treatments were incubated at 26 ± 2 °C and exposed to a 16 h light/day photoperiod under constant fluorescent light of 1500 Lux in the growth chamber. After 4 weeks of culturing on media, the following data were recorded: 1—Average callus fresh weight (g), 2—Average callus dry weight (g).

#### 4.2.2. Callus Crude Extraction

The obtained callus was dried, and ground to fine powder by a mortar and pestle. Five grams of dried powdered callus of each treatment was extracted for roughly 6 h at 60 °C using a Soxhlet apparatus with 50 mL of 100% methanol. The solution was then evaporated to dryness at 40 °C using a rotary evaporator [51]. Next, the crudes were stored in glass bottles at −20 °C for further bioassays.

#### 4.2.3. The GC–MS Analysis

The GC–MS analysis of methanolic crude extracts was conducted for the identification and characterization of various chemical components, as well as the presentation of the total extract from samples. A Thermo Scientific Trace GC Ultra/ISQ Single Quadrupole MS (Thermo Scientific, Waltham, MA, USA) TG-5MS-fused silica capillary column was used for the GC–MS study (30 m, 0.251 mm, 0.1 mm film thickness). An electron ionization system with 70 eV ionization energy was employed for GC–MS detection, with Helium as the carrier gas at a constant flow rate of 1 mL/min. The temperature of the injector and MS transfer line was fixed to 280 °C. The oven temperature was set to rise from 50 °C (hold for 2 min) to 150° C at a pace of 7 °C per min, then to 270 °C at a pace of 5 °C per min (hold for 2 min), then to 310 °C at a rate of 3.5 °C per min as a final temperature (hold 10 min). A percent relative peak area was used to evaluate the quantification of all the discovered components. The chemicals were tentatively identified by comparing their respective retention times and mass spectra to those of the NIST, WILLY library data from the GC–MS instrument.

### 4.3. In Vitro Anti-Cancer Study (Cell Line Studies)

Following the method of Etman et al., 2020 [52], human colorectal adenocarcinoma cell line (Caco-2) (ATCC^®^ HTB-37™) was employed in this study and all experiments were carried out in CERRMA (Center of Excellence for Research in Regenerative Medicine and its Applications), Faculty of Medicine, Alexandria University. Cells were cultured in Dulbecco’s modified eagle medium (DMEM)-high glucose enriched with (10% *v*/*v*) fetal bovine serum (FBS) and antibiotics (100 U/mL penicillin, 100 μg/mL streptomycin). Cells were incubated in 5% CO_2_ at 37 °C for maintenance and media was changed every 2 days. Cells were passaged on reaching 80–90% confluence using 0.25% (*w*/*v*) trypsin/ethylene diamine tetra acetic acid (EDTA), then plated in T75 cm^2^ flasks or in other culture vessels (6 or 96 well plate) according to the experiment conducted.

#### 4.3.1. Cellular Cytotoxicity Assay

Different callus extracts were dissolved in 1 mL dimethyl sulfoxide (DMSO) to assess their cytotoxicity through applying different concentrations, in which the highest concentration of DMSO would never exceed 0.1% DMSO to avoid its cytotoxic effect. Cellular cytotoxicity was assessed using MTT assay as described by El-Habashy et al., 2020 [53]. Caco-2 cells were seeded at a density of (5 × 10^3^) in 96-well plate. Each well contained 100 μL of culture media. Cells were allowed to adhere for 24 h. Then, they were treated with different concentrations ranging from 10 to 200 ug/mL or 100 to 350 ug/mL of the LAA-, CA- and control-treated groups, respectively, then incubated for 48 h. After incubation, 100 μL of fresh media containing a 10 μL MTT solution (5 mg/mL) was added and incubated for another 4 h in a CO_2_ incubator. Finally, 100 μL DMSO was added to dissolve the produced formazan crystals. Absorbance was measured at 570 nm using a microplate reader. The viability of cells was determined according to the following equation: % viability = Absorbance of sample at 570 nm/Absorbance of untreated at 570 nm × 100, where untreated cells were treated with culture media only. The effect of different concentrations on cell viability was expressed as % inhibition against concentration and used to calculate IC50 (concentration required to kill 50% of the cells). Results were expressed as mean ± SD (*n* = 8).

#### 4.3.2. Colony-Forming Assay

Clonogenic assay is an in vitro cell-survival assay based on the ability of a single cell to grow into a colony. The assay tests the cell for its ability to undergo division with and without treatment, thus, used to determine the effectiveness of cytotoxicity. The assay was performed by plating 10^3^ cells on a 60 mm culture dish in CCM; after 24 h, the media -was discarded from the wells and replaced with culture media only or media with added treatment of half-calculated IC50 of the LAA, CA and control for 14 days. The medium was changed every 2–3 days; then, after 14 days, cells were washed with PBS, fixed and stained using Crystal Violet (Sigma-Aldrich, Burlington, MA, USA) at 3% (*w*/*v*). The number of visible colonies was counted and plating efficiency, (the number of colonies formed/number of cells plated) × 100 was calculated for comparison.

#### 4.3.3. Ki-67 Flow Cytometry Proliferation Assay

Caco-2 cells were plated in 6-well plates at a density of 2 × 10^4^ cells per well for the cell proliferation assay. After incubation for 24 h for adherence, the media was discarded from the wells and replaced with culture media only or media with added treatment of half-calculated IC50 of LAA, CA and control groups for 48 h. The culture medium was then removed from the wells, cells were washed twice with sterile PBS, and 0.25% of the trypsin-EDTA solution was added to detach the cells. After 5 min of incubation at 37 °C and 5% CO_2_, fresh culture medium was added to inactivate trypsin, and cells were collected in flow cytometry tubes. Cells were then labelled using the Ki67 Proliferation Kit (D3B5, Rabbit mAb Alexa Fluor^®^ 488 Conjugate, Cell signaling technology) according to the manufacturer’s instructions (cell signaling technology; flow cytometry, methanol permeabilization protocol) and analyzed by using BD FACS Calibur flow cytometry. Unstained control cells were used for gating to determine the percentage of proliferation of the Ki67-positive cells in the samples. Percentages of proliferating (Ki67 positive) cells were used to calculate the means ± SD for each group in triplicate [54].

#### 4.3.4. Migration Assay (Scratch Assay)

The scratch assay was carried out to test the ability of the LAA, CA and control groups to attenuate the migration of cancer cells [55]. The cells were grown until 70–80% confluency in complete media. Then, they were incubated for 24 h in a 6-well plate to allow cellular adhesion. Next, media was removed and replaced with starvation (serum-free) medium for another 24 h. Then, a scratch was carried out using a sterile 200 μL pipette tip and cells were washed with PBS twice to remove any detached cells; then, the cells were treated with half-calculated IC50 in the LAA, CA and control groups in starvation media, or just starvation medium in the control–untreated group, for 48 h. Images of the scratch were taken using an inverted phase contrast microscope (Olympus, Waltham, MA, USA) after scratching and this was marked as zero time. Images were then taken after 24 and 48 h. Representative images were taken, and the area of wound-healing was calculated using Image J software (Version 1.52p software 32, NIH, USA).

### 4.4. Statistical Analysis

GraphPad Prism 8 was used for statistical analysis (GraphPad Software, La Jolla, CA, USA). The data were analyzed using one-way analysis of variance (ANOVA). Significant differences were determined to have *p*-values less than 0.05.

## 5. Conclusions

In the current study, we found that adding L-ascorbic acid to media enhanced the secondary metabolite production in *R. damascena* callus and it has an efficient antiproliferative, anti-clonogenic and anti-migratory effect on Caco-2 cancer cells, thus, can be used as an adjuvant anti-cancer therapy.

## Figures and Tables

**Figure 1 molecules-27-06241-f001:**
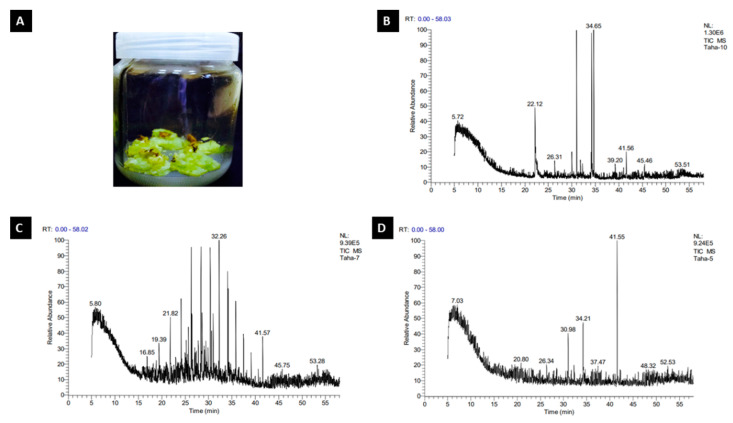
Callus adequate mass (**A**); GC–MS chromatograms of callus elicited by control (**B**); citric acid (**C**); L–ascorbic acid (**D**).

**Figure 2 molecules-27-06241-f002:**
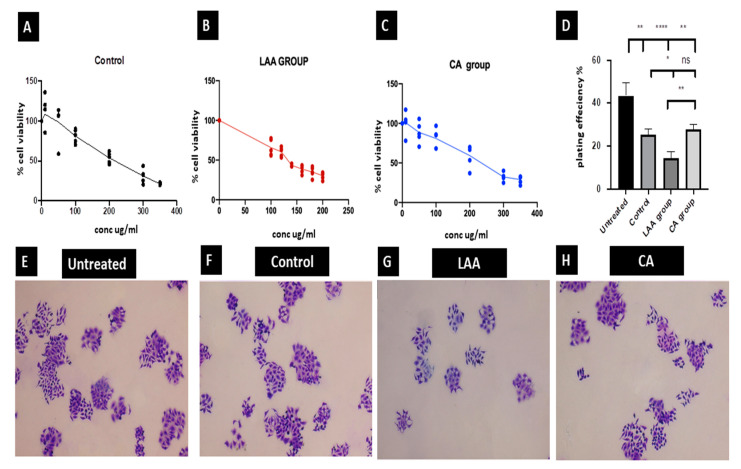
Cellular cytotoxicity (**A**–**C**); clonogenic assay statistical analysis of plating efficiency (PE) of tested groups blotted at the *X* axis, PE % = the number of colonies formed/number of cells plated) × 100 blotted at the Y axis, the untreated group showed the highest PE while the LAA group showed the least PE, being significantly lower than the CA, control and untreated groups. * Significance at *p* < 0.05, ** Significance at *p* < 0.001, **** Significance at *p* < 0.0001, ns No significant difference. (**D**) Clonogenic assay images showing cell colonies stained with crystal violet imaged using inverted light microscope X40 (Olympus CKX41SF, Tokyo, Japan) (**E**–**H**). Untreated (**E**); control (**A**,**F**); L-ascorbic acid (**B**,**G**); citric acid (**C**,**H**).

**Figure 3 molecules-27-06241-f003:**
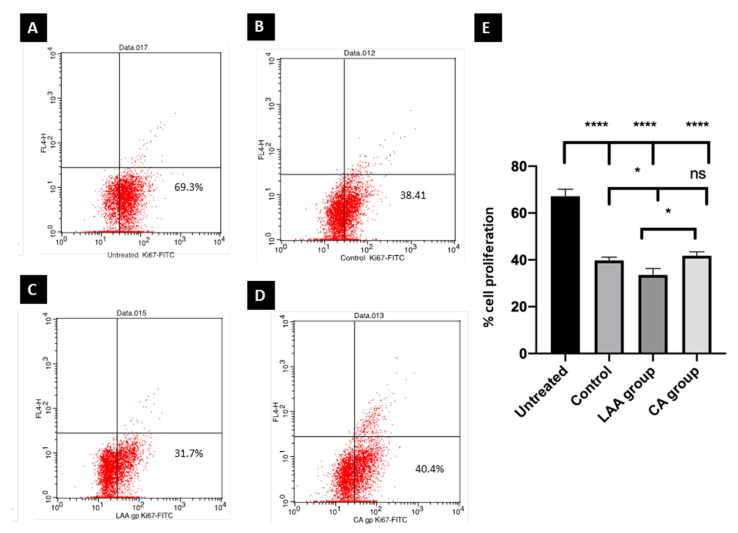
Ki-67 flow cytometry proliferation assay. Untreated (**A**); control (**B**); L-ascorbic acid (**C**); citric acid (**D**). Statistical analysis for the cell proliferation of different groups blotted on X axis and % of proliferation on Y axis, where proliferation of cells in all groups was significantly lower than the untreated group. **** Significance at *p* < 0.0001, LAA group showed the lowest proliferation compared to the crude callus-treated (control) and CA-treated groups. * Significance at *p* < 0.05; there was no significant difference between the control and CA groups (ns) (**E**).

**Figure 4 molecules-27-06241-f004:**
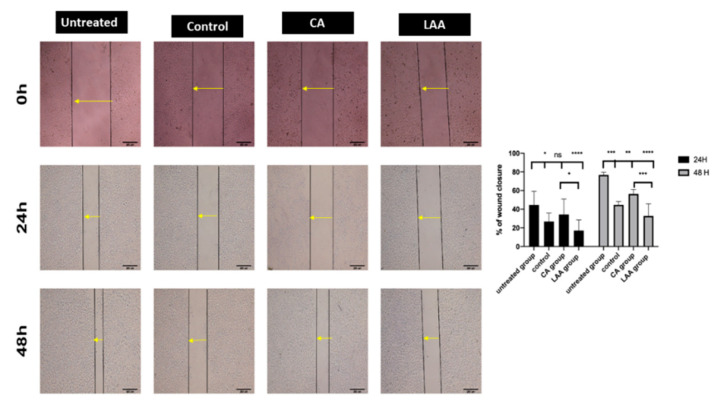
Migration assay at 0, 24 and 48 h. Statistical analysis for the % of wound closure of different groups at 24 and 48 h. * Significance at *p* < 0.05, ** significance at *p* < 0.001, *** significance at *p* = 0.0001, **** significance at *p* < 0.0001.

**Table 1 molecules-27-06241-t001:** The effect of bio-elicitors on *R. damascena* callus development and total crude weight.

Treatments	^1^ Fresh Weight (g)	Dry Weight (g)	Crude Weight (g)
Control	^2^ 2.166 ab	0.214 b	0.038 a
Citric acid 0.5 g/L	1.267 b	0.168 b	0.022 a
L-ascorbic acid 0.5 g/L	2.878 a	0.306 a	0.032 a
L.S.D at 5%	1.019	0.075	2922.8

^1^ Each treatment was represented by ten replicates, each with three explants. ^2^ Means with different letters within the same column or row differ significantly (*p* < 0.05).

**Table 2 molecules-27-06241-t002:** GC–MS analysis of *R. damascena* callus in the control treatment.

Peak	R. Time	Area	Area %	Name
1	5.72	2561	1.04	(2,3-Dihydro-5,10,15,20- tetraphenyl [2-(2)H1]prop hyinato) copper(II)
2	22.12	2251	9.15	Benzonitrile (CAS)
3	26.31	3938	1.60	1-Pentanol, 2,2-dimethyl-(CAS)
4	34.65	3385	13.75	Octadecanoic acid, methyl ester (CAS)
5	39.20	3807	1.55	4-Octanol, propanoate
6	41.56	7511	3.05	1,2-Benzenedicarboxylic acid, bis(2-ethylhexyl) ester (CAS)
7	45.46	4280	1.74	2,6-Nonadien-1-ol
8	53.51	2360	0.96	(5,10,15,20-Tetraphenyl [2-(2)H1]prophyrinato)zinx(II)

**Table 3 molecules-27-06241-t003:** GC–MS analysis of *R. damascena* callus in the citric acid treatment.

Peak	R. Time	Area	Area %	Name
1	5.80	3614	0.95	Dichloroacetaldehyde
2	16.85	4391	1.16	Dodecane, 5,8-diethyl-(CAS)
3	19.39	7554	2.00	Tetradecane (CAS)
4	21.82	1355	3.58	Tetratetracontane (CAS)
5	32.26	2202	5.82	Octadecane, 2-methyl-
6	41.57	1018	2.69	1,2-Benzenedicarboxylic acid, diisooctyl ester (CAS)
7	45.75	2866	0.76	3,3′,5,5′-Tetrabromo-2-nitro-2′-propylsulfonylbiphenyl
8	53.28	2764	0.73	5á-Pregnan-20-one, 3à,11á,17,21-tetrakis(trim ethylsiloxy)-, O-methyloxime

**Table 4 molecules-27-06241-t004:** GC–MS analysis of *R. damascena* callus in the L-ascorbic acid treatment.

Peak	R. Time	Area	Area %	Name
1	7.03	2904	1.79	11-[(t-Butyldimethylsilyl)oxy]-6,9a-dimethyl-6-(m ethoxycarbonyl)-(perhydro )napthaleno[a]benzofulvene
2	20.80	1980	1.22	Mixture of: 5,6-Dihydro-6-methy l-2H-pyran-2-one and 5-methoxy-3-pente ne-2-ol
3	26.34	3210	1.98	Nonane, 1-chloro-(CAS)
4	30.98	9453	5.82	Hexadecanoic acid, methyl ester (CAS)
5	34.21	1391	8.57	7-Octadecenoic acid, methyl ester (CAS)
6	37.47	1777	1.09	(2,3-Dihydro-5,10,15,20-tetraphenyl [2-(2)H1]prophyrinato)copper(II)
7	41.55	2884	17.75	1,2-Benzenedicarboxylic acid, mono(2-ethylhexyl) ester
8	48.32	1799	1.11	6-[2′-(4″-Phenyl)ethyl]-1,2,3-triphenyl-9H-tribenzo[a,c,e]cycloheptatriene

## Data Availability

Not applicable.

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
