# Peer review of "Evaluation of the Anti-Cancer Potential of Rosa damascena Mill. Callus Extracts against the Human Colorectal Adenocarcinoma Cell Line"

_molecules, 2022, doi:10.3390/molecules27196241_

Round 1
Reviewer 1 Report
Dear Authors,
the manuscript "Evaluation of the anticancer potential of Rosa damascena Mill. callus extracts against human colorectal adenocarcinoma cell line" describes the influence of ascorbic or citric acid on callus growth parameters, secondary metabolites content and cytotoxic activity of the plant extracts. The manuscript has merit, however there are some points that need explanation/corrections:
1. the IC50 values should be write in the text as ug/ml, not only "ug".
2. The Authors show in the manuscript the tables with the content of secondary metabolites in each extract (control, LAA and CA) and concluded that the added acids enhanced production of secondary metabolites in Rosa callus. In those tables most of described metabolites are different, so how to compare the production of these compounds among the control, LAA and CA? The total weight is actually higher, but what about the content of metabolites?
3. In MTT cytotoxicity assay, the Authors showed the IC50 values of control, LAA and CA extracts (180.6, 137.8, 237), however the results presented on graph (Fig. 2) indicate other value for control - this is about 100 ug/ml, so the control is the most active. Also, what is on Fig. 2 D? the y axis should be described. The figure caption should be corrected (control should be A,F; LAA should be B, G, CA should be C, H, etc.) and described in details (e.g. what is the control in the experiment?; how many times the experiment was repeated). There is also double "f" on the figure.
4. Fig. 3E - the y axis should be described. Also figure caption should have more details.
5. Fig. 4 - the scratch size should be the same in 0 h. Fig. 2 indicates that in the untreated probe this size is much bigger at the beginning. Was the scratch made after incubation or before? (line 378).
5. Cytotoxicity assays - some control, non-cancer cell line should be added in the experiments to show potential differences. Also, in what kind of solvent did the Authors dissolve dry extracts? What was the maximum concentration of the solvent which the cells were treated?
6. In MTT assay - did the Authors test the potential interaction between the MTT reagent and extracts to avoid false results?
Minor comments:
7. line 29 - "are" should be converted to "were".
8. latin name of Rosa damascena should be write in italics in whole text. Also, the Authors wrote R. damascena once, once R. damascina.
9. line 187 - should be CA, not CAA.
10. the name of cell line Caco-2 should be write the same in the text.
11. line 139 - what is "ne"?
Author Response
Dear Sir,
Thank you for your valuable comments. The authors did all of required corrections.

Reviewer 2 Report
The manuscript ‘Evaluation of the anticancer potential of Rosa damascena Mill. callus extracts against Human colorectal adenocarcinoma cell line’ highlighted the importance of adding ascorbic acid as elicitor in the culture media of Rosa damascene to improve its content on secondary metabolites as well as it cytotoxic effects. However, this work necessitate some improvements:
Abstract should be no more than 200 words
Figure 2: the cytotoxicity for the untreated cells is not present. Please revise the figure and the legend
Figure 4. By comparing the results for the control and the LAA after 48h (please see the photo), it seems that the control may have a lesser closure percentage compared to LAA. Could you please re-verify this ?
More information regarding the extraction method is needed, as residual ascorbic acid may give biased results.
An Lc-MS analysis is missing, as volatile compounds would represent only a small fraction of a methanolic extract.
A simple dosage of flavonoids or polyphenols could help confirming the ability of ascorbic acid increasing the secondary metabolites content.
Line 45-46 : “In 2020, 104 and 610 new cases of colon cancer were estimated and 45 576 and 858 people died from it”. The sentence is not very clear, could you please rephrase it.
Line 72-73. Please re-consider the classes of secondary metabolites. Please refer to the introduction part of this paper: https://doi.org/10.4103/jrptps.jrptps_6_18
Line 85: “anti-HIV effects’’. Please add a reference
Line 86: Add a reference
Line 90: in vitro must be in italic form. Please revise the whole document
Line 101-102: Please remove the parentheses and rephrase
Lines 116, 120, 132: Please put R. Damascina in italic form
Line 122-125: Please rephrase
134-135: Please rephrase
Line 225-230: please rephrase
Line 232: change ‘foliar’ by ‘foliar application of
Author Response

(The authors gave the same response as above.)

Round 2
Reviewer 1 Report
Dear Authors,
most of my comments have been done in the manuscript, however points which are below need corrections:
1. The y axis on Fig. 2D should be described (because readers will not know what is the value 0.2; 0.4, etc. (CFU? need explanation).
2. The same situation is on Fig. 3E. The y axis should have the name: percentage of proliferation of the cells.
3. The maximal concentation of DMSO (0.1%) added to the cells should be added in Materials and Method section.
Author Response
Dear Sir,
Thank you again for your valuable comments. All of your required comments have been done in the manuscript.

Reviewer 2 Report
The authors have made important changes to improve the quality of the manuscript. However, there are still a few points to clarify:
Line 40: Please pay attention to the statistics. Again, please refer to the article where you got this data from and reconsider this statement as the numbers you gave do not match the original article.
Line 68-93: Please add the reference from which you based to modify your introduction https://doi.org/10.4103/jrptps.jrptps_6_18.
You should explain why you chose GC-MS analysis instead of LC-MS. In your previous answer there is a misunderstanding and I could not catch the meaning of your explanation. LC-MS is NOT a specialized tool for identifying volatile chemicals in samples.
Author Response

(The authors gave the same response as above.)
